# Total Costs of Centralized and Decentralized Inventory Strategies—Including External Costs

**Dariusz Milewski**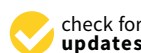

Department of Organization and Management, Institute of Management, Faculty of Economics,
Finance and Management, University of Szczecin, 71-004 Szczecin, Poland; Dariusz.Milewski@usz.edu.pl

**Abstract:** The paper deals with the economic efficiency of decentralized and centralized strategies of distribution goods in terms of both internal efficiency of firms and external costs of logistics processes (first of all external costs of transport). The author developed a model (using an electronic spreadsheet) in order to calculate the economical efficiency in the micro and macro dimensions in order to find the distances on which distribution using one central warehouse is more profitable than decentralized distribution. The results of the simulations show that the strategy of centralized inventories can be in many cases an economically effective strategy although not for deliveries on very long distance. The results confirm that the benefits of centralization are lower inventories, although the simulations do not confirm the applicability of the square root law to calculate the level of inventories. However, they confirm a positive impact on the level of logistic customer service, measured by the availability of stocks. Better service is probably the main benefit of this strategy. In order to investigate the impact of individual parameters on the total costs of logistics processes 1300 simulations were carried out for various cases: The volume of annual sales, fluctuations in demand, the value of distributed goods, the number of warehouses in a decentralized system and the width of the product range, costs of warehousing, and maintaining stocks, and the distance of transport and in deliveries to customers.

**Keywords:** centralization of distribution; "square root law" of inventory management; economical efficiency of logistics processes; simulation; external costs of transport

## 1. Introduction

The classic trade-off problem between the costs of transport, storage, and inventory maintenance and lost sales is associated with the choice of a centralization or decentralization strategy of distribution. It affects not only the costs of logistics processes, but also the level of logistics services, and thus the competitiveness of the enterprise. It may also have an impact on external costs, although as it will be shown later in the article, the impact is not clear.

The centralized inventory strategy is used by some companies. The benefits of this strategy are primarily the reduction of inventories and warehousing costs. As deliveries in this strategy have to be made sometimes over longer distances and possibly directly to customers, transportation costs may increase; especially if, after the centralization, goods are delivered in small quantities. Transportation costs may also increase due to the need to increase the speed and flexibility of deliveries. Usually, the implementation of the strategy of centralized inventories in distribution contributes to the improvement of the level of logistics service measured by the inventory availability. Although the distance is increasing and thus the delivery time and there is a risk of delays in deliveries, the availability of stock in the central warehouse is usually greater.

The centralization strategy also has consequences for the environment. Increase of develieries in smaller quantities can not only result of higher operational costs of transport, but also higher external costs of transport.

There are known cases of companies that have successfully implemented a strategy of centralizing inventory in distribution. However, there is a lack of knowledge about the efficiency factors of this strategy such as types of goods, demand characteristics, and costs of logistics processes.

The effects of this strategy may vary, and the reasons for centralizing inventories may vary. Some companies wanted to reduce inventory and warehousing costs, others to improve service levels. In some cases, transportation costs have increased, total logistics costs have been reduced, and the level of customer service has improved. Transport costs (including external costs), even if road transport is used, do not have to increase significantly and may even decrease if distribution is taken over by a specialized logistics operator. Direct delivery distances in centralized distribution may also be shorter in some cases.

The aim of this paper is to present possible impact of the implementation of the strategy of centralized inventory in distribution on the economic efficiency of enterprises and external costs of transport. With the use the model elaborated by the author, analyses have been conducted, in order to identify impact of different factors of this efficiency, the results of which are presented in the following parts of the paper.

## 2. Literature Review

Although the strategy of centralization of inventories is quite popular, its impact on the economic efficiency of the enterprise and external costs is not the subject of much scientific research.

The relationship between the number of warehouses in the distribution network and the level of stocks is included in the so-called the "square root" formula. Assuming that the level of service measured by the availability of inventory is constant, the inventory decreases when the number of warehouses in the distribution network decreases. It should be noted, however, that this formula allows for the calculation of only a part of stocks, namely the safety stock [1,2].

The usefulness of this formula in practice is quite often subjected to criticism also in the scientific literature. The results of empirical research conducted in various sectors of the economy are very diverse. For example, according to the research by Oeser and Romano, the application of this formula is very limited, and the most similar results were obtained rather in trading companies, than in the production ones. [3]. Moreover, the savings resulting from centralization may be greatly overestimated [4]. Some research authors suggest that safety stocks and cyclical stocks should be considered together. For example, Das believes that the formula applies, but the cyclical stock should also be considered. [5]. B. Fleischmann also points out the need to take into account the volume of deliveries to warehouses [6], and that the formula does not apply if deliveries are made to these warehouses in full truck loads.

The type of business is a very important factor in the efficiency of a distribution strategy. The centralized inventory strategy may be effective in industries where the costs of maintaining inventory and their storage are very high, e.g., in the food industry [7].

Another important factor enabling the effective implementation of this strategy is transport—the costs and quality of transport services [8]. Examples of companies implementing this strategy confirm this. Its effectiveness often depends on cooperation with transport or logistics operators who are able to effectively deliver goods directly from central warehouses to final receivers of goods.

The problem that arises here is the increase in transport costs. These are costs not only incurred by enterprises, but also by society as a whole in the form of higher external costs of transport. However, according to some authors, reducing the number of wareshouses can also help in minimizing external costs by reducing the consumption of energy and other resources [9]. The factor contributing to such positive effects is digitization, which is a substitute for physical processes and material consumption—e.g., paper documents. An alternative to physical warehouse facilities may also be virtual warehouses. Moreover, it should be borne in mind that the processes of movement are also carried out in warehouses and that warehouses use energy (lighting, heating).

The problems of optimization of logistics processes in connection with production processes has been discussed many times in the research of the author of this publication. Overall, the conclusions that can be drawn from these studies are as follows [10,11]:

- Optimization of logistics processes can in many cases improve the profitability of enterprises, even if the share of costs of logistics processes in total costs is not high. The impact of this optimization depends on various factors, including on the value of the goods.
- An optimal level of logistic customer service very often has a greater impact on the economic efficiency of enterprises than just the optimization of logistics costs.
- Optimizing using optimization formulas in many cases is too simplified and may not give correct results. The simulation method using models can be more effective.
- Like the authors cited earlier, the author of the paper also thinks that there are relationships between the delivery sizes, the level of the safety stocks, and the level of customer service, therefore these parameters should be considered together.

The problem of the impact of transport solutions on the external costs of transport has already been the subject of the author's research, who analyzed the possibility of reducing external costs of transport by increasing the efficiency of processes in road transport [12]. They were carried out for different variants of mileage utilization in road transport, different levels of external costs, using different methodologies. The results show that despite better efficiency of transport means, the rail transport is in all cases greener. Therefore, to justify the use of road transport, other costs of logistics processes should be taken into account.

However, there is lack of the research that would capture the problem of economic efficiency of centralized inventory (in terms of micro and macro) in a systemic manner, supported by concrete calculations. The results of such research are presented later in the paper.

## 3. Materials and Methods

The main method used in the research is the method of the computer simulation using the model developed by the author. The model generates daily sales, logistics processes for a whole year, and makes calculations. The tool used in the simulation is the Excell spreadsheet, which has a function RAND(). In the model, daily demand is generated with a given deviation. The assumption is that demand is normally distributed.

Simulations have been conducted for the following variants of the two strategies:

- Decentralized—6 local warehouses (Distribution Centers).
- Centralized—1 central warehouse.

The warehouses in both strategies are supplied directly from the production plant. From the warehouses in both strategies, goods (12 products) are delivered directly to customers (e.g., shops). The assumption is that the distribution in the decentralized system from the local warehouses to customers ("last mile") is performed over distances of 50 km on average (Figure 1).

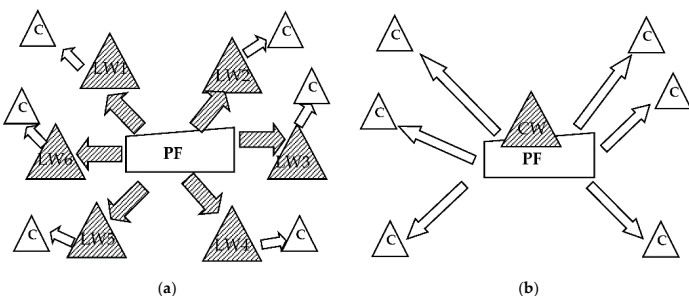

(a)                 (b)

**Figure 1.** Decentralized and Centralized systems of distribution (**a**) Decentralized system; (**b**) Centralized System. PF—production facility; LW—a local warehouse; CW—a central warehouse; C—customer.

In both strategies, the flow of finished products from the production facility is "pushed" to the warehouses. Deliveries to final receivers from these warehouses are "pulled" according to the actual orders of customers.

With the use of this model, a number of simulations were conducted based on optimization criteria—profit margin and total costs of deliveries. These costs consist of the costs of logistics processes in distribution and the costs of lost sales, which are not included in traditional accounting.

The annual total cost of distribution consists of the following components:

- Reloading costs in the production plant and warehouses ($C_r$);
- Warehousing costs ($C_w$);
- Inventory costs ($C_i$);
- Transport costs to warehouses and to customers ($C_t$);
- The costs of lost sales ($C_{ls}$).

These costs are calculated as follows:

$C_r$ = Unit cost of reloading * Sales volume,

$C_w$ = Unit cost of warehousing * Max inventory level,

$C_i$ = Unit cost of production * Average inventory * Cost of maintaining inventories,

$C_t$ = Freight rate * Distance * Sales volume,

$C_{ls}$ = (100%-Level of Service) * Sales volume * Sales price.

At all stages (with the exception of loading and warehousing processes in the production plant and central warehouse), the company uses transport and warehousing services. The rates used in the simulations are based on the actual rates on the Polish logistics services market.

To calculate the costs of transport (internal and external) and storage costs, the parameters of the shipments (number of pallets and loading weight) are needed. It was assumed that road vehicles with a net capacity of 24 tons and EUR 34 pallets are used.

It was also assumed that the company rents warehousing space and pays a fixed rate. Thus, unlike the costs of maintaining stocks, which are calculated on the base of average level of stocks and their value, the warehousing costs are affected by their maximum level, regardless of the value.

The unit cost of storage and handling is lower in the case of the central warehouse, because it was assumed that it is more efficient than local warehouses.

In the decentralized strategy, there are two variants of deliveries to the local warehouses:

1. Deliveries by rail on a schedule every week to each warehouse.
2. Deliveries by road (24 tons—like in the centralized system) daily for each warehouse.

Deliveries to customers are made on the basis of actual orders but taking into account the availability of goods in local warehouses in the decentralized system and central warehouse in the centralized system.

The data for the simulation are presented in Table 1.

The analyses assumed that the prices for the service in rail transport are by 40% lower, which seems to be a reasonable assumption. The actual price may be even lower—for example, for large customers for whom transport is carried out regularly, in large batches (e.g., 3000 tons), and a contract lasting several years is signed with the customer.

The simulations were performed in order analyze the impact of both strategies on:

1. The level of inventories and logistics customer service;
2. The company's profitability;
3. The internal costs total costs of distribution;
4. The total external costs of transport.

**Table 1.** Assumption for calculations.

| Yearly Sales | | | |
|---|---|---|---|
| [tonnes/year] | | [pellets/year] | |
| **240,000** | | **340,000** | |
| **Low Value Goods** | | **High Value Goods** | |
| Unit production costs [PLN/tonne] | Selling price [PLN/tonne] | Unit production costs [PLN/tonne] | Selling price [PLN/tonne] |
| 2000 | 2300 | 20,000 | 22,500 |
| **Logistics costs** | **Distribution Strategies** | | |
| | **Weekly deliveries** | **Daily deliveries** | **Central Warehouse** |
| Unit costs of reloading [PLN/pellet] | 3 | 3 | 2 |
| Capital costs of inv. | 13% | 13% | 10% |
| Costs of warehousing [PLN/pellet/month] | 35 | 35 | 25 |
| Transport costs [PLN/tonne/km] | 0.10 | 0.16 | 0.16 |

Source: Own calculations.

The calculations of the total costs of distribution (point 3) were conducted first for two cases and then 1300 simulations for different variants results for which were the base for drawing general conclusions.

Simulations 3 and 4 were intended to calculate (with the adopted assumptions) at which distances the centralization strategy is more profitable than the decentralized.

All simulations were carried out for two types of goods: Low and high value. In practice, the value of the goods transported can vary greatly. For example, in the case of agricultural products, a trailer may contain goods worth PLN 55,000. In the case of electronics, it can be over PLN 10,000,000. Simulations have been conducted for goods of value of 2000 PLN/ton and 20,000 PLN/ton.

## 4. Results

### 4.1. The Impact of Centralization of Warehousing on the Level of Inventories and the Customer Service

In the first stage, the impact of these distribution strategies on the level of logistic customer service was calculated measured by the availability of inventory in the warehouses from which deliveries to customers are made. To ensure comparability, the simulation was performed for the same levels of safety stocks in all cases.

The simulation results are presented in Table 2 and Figure 2. They seem to confirm the positive impact of centralization of inventories on the level of service.

**Table 2.** Impact of Demand Volatility on the Level of Service at the Central and Local Level.

| Strategy | Standard Deviation of Average Demand | | | | |
|---|---|---|---|---|---|
| | **2.2** | **2.4** | **3.4** | **3.9** | **4.5** |
| | [t] | | | | |
| | **Customer Service Level** | | | | |
| Weekly deliveries | 99.88% | 99.71% | 99.52% | 99.35% | 98.98% |
| Daily deliveries | 99.87% | 98.16% | 97.04% | 92.70% | 91.95% |
| Central Warehouse | 99.87% | 99.74% | 99.88% | 99.54% | 99.42% |

Source: Own calculations.

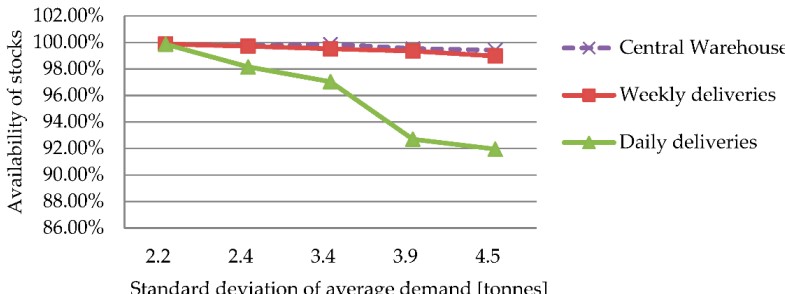

**Figure 2.** Impact of Distribution Strategies on Customer Service (availability of stocks).

With a low standard deviation, the level of service is similar in all cases. However, it decreases when changes in demand become larger, but at a different pace. The greatest impact is seen in the decentralized strategy with daily deliveries. In the centralized system, the availability of stocks is the highest and remains relatively high. It may seem, then, that in fact, according to the "square root law", the same level of service can be maintained in a centralized system with a lower level of inventory.

It may come as a surprise that the levels of services for weekly deliveries in the decentralized system and centralized system were similar. However, this confirms the results of analyzes carried out for years by the author of this paper, that for higher quantities of deliveries, the level of service may be also higher [10,11]. It also means that with larger deliveries, the level of safety stocks may be lower while maintaining the same level of service. Deliveries to the warehouse in larger quantities, but less frequently, reduce the risk of running out of stock.

Since a similar level of service can be obtained for deliveries by rail transport to local warehouses and deliveries directly from the central warehouse, it may be assumed that maintaining a network of local warehouses with daily deliveries in small quantities is pointless. This is confirmed by further analyzes.

Then, simulations were carried out to check whether there were differences between the results obtained by applying the "square root" formula and the simulation model. The results of this simulation are presented in Table 3.

**Table 3.** Differences of Levels of Safety Stocks From Simulations and "Square Root" Formula.

| Standard Deviation [tons] | 2.18 | | | | | |
|---|---|---|---|---|---|---|
| Customer Service | 97.5% | 98.0% | 98.5% | 99.0% | 99.5% | 100.0% |
| Weekly deliveries | 10% | −5% | 1% | −16% | −42% | −58% |
| Daily deliveries | −75% | −175% | −246% | −315% | −402% | −348% |
| **Standard Deviation [tons]** | **2.81** | | | | | |
| Weekly deliveries | 24% | 8% | 2% | −13% | −23% | −16% |
| Daily deliveries | 7% | −22% | −48% | −90% | −120% | −152% |
| **Standard Deviation [tons]** | **4.44** | | | | | |
| Weekly deliveries | 30% | 18% | 5% | −10% | −30% | −32% |
| Daily deliveries | 30% | 12% | −2% | −24% | −46% | −53% |

Source: Own calculations.

The data in table show the differences between the level of safety stocks calculated with use the simulation method (proposed by the author of the paper) and the safety stocks calculated using the "square root" formula.

The differences are very large and depend on the strategy used, the degree of volatility in demand, and the service level. This means that it would be difficult to modify this formula to reflect the actual impact of centralization on inventory levels. The relatively closest results are for service levels between 98% and 99.5% and, perhaps surprising again, deliveries in larger batches (weekly by rail). Perhaps this formula was developed just for such situations, when there is high volatility of demand, the required level of service is high, and deliveries to local warehouses are performed in relatively bigger quantities.

Moreover, the question arises here: What is the usefulness of this formula, taking into account that it only applies to the safety stock, which (as shown earlier) should be established together with a delivery quantity? In addition, while it is quite easy to calculate the costs directly related to their average level, i.e., the costs of e.g., capital tied up in inventories, the costs of their warehousing also depend on other factors. The cost of storing them in a central warehouse may be lower not only because their level in this warehouse will be lower, but also due to greater efficiency resulting from economies of scale (better use of the space of a large warehouse, greater efficiency of a large warehouse, where it is profitable to use more efficient warehouse technologies)

### 4.2. Impact of the Distribution Strategy on the Company's Profitability

Table 4 shows the structure of the costs of manufacturing and distribution of products, for the two strategies (three variants) and the two values of goods (cheaper and expensive) for the case where the standard deviation of the average sales is high and the service level is 97.5%.

**Table 4.** Structure of Costs in Different Distribution Strategies (Standard Deviation 4.44 t, Customer Service 97.5%).

| Strategy | Weekly Deliveries | | Daily Deliveries | | Central Warehouse | |
| Goods | Cheaper | Expensive | Cheaper | Expensive | Cheaper | Expensive |
|---|---|---|---|---|---|---|
| Total production costs [PLN/year] | 82.56% | 86.00% | 82.54% | 85.98% | 82.88% | 86.33% |
| Reloading costs [PLN/year] | 0.53% | 0.05% | 0.53% | 0.05% | 0.11% | 0.02% |
| Yearly transport costs [PLN/year] | 2.33% | 0.24% | 3.50% | 0.36% | 3.37% | 0.35% |
| Yearly warehousing costs [PLN/year] | 1.03% | 0.11% | 0.86% | 0.09% | 0.39% | 0.04% |
| Yearly inventory costs [PLN/year] | 0.22% | 0.23% | 0.22% | 0.23% | 0.06% | 0.06% |
| Lost sales | 2.61% | 2.61% | 2.58% | 2.58% | 2.59% | 2.63% |

Source: Own calculations.

With these assumptions, production costs dominate the total costs, which may raise the question of whether the optimization of logistics processes in distribution is in this case important from the point of view of the company's economic efficiency? However, even with a small share of the distribution costs, the optimization of logistics processes in distribution may have a significant impact on a profit margin.

The largest share in the structure of distribution costs in all cases are the costs of lost sales, even in the case of the cheaper goods. This confirms the great importance of logistics services identified in the literature. Such conclusions also result from the author's previous research [10,11]. It can therefore be assumed that this factor is of greatest importance for the effectiveness of the distribution strategy. This will be verified later.

The share of other costs depends on the value of goods—in the case of more expensive goods, their share is small. In second place, however, after the cost of lost sales, there are transport costs. Therefore, the transport distances are important. The simulation results concerning the impact of the transport distances and the levels of service on the profitability are presented in Table 5 for cheaper goods, and for more expensive ones, Table 6.

The results are interesting, although in line with expectations, i.e., in the case of cheaper goods, the greatest impact on the margins is the distance of transport, and in the case of more expensive ones, the level of customer service. These results are not difficult to interpret—in the case of cheaper goods, transport costs have a greater share, and the more expensive ones, costs of the lost sales. It is also worth paying attention to an interesting relationship. In the case of cheaper goods, a high, but not the highest, level of service is the best. However, in the case of expensive goods, the level of customer service should be very high.

**Table 5.** Margins in Different Strategies (Lower Value of Goods, Standard Deviation = 4.44 t).

| Distances [km] | Customer Service | | | | | |
|---|---|---|---|---|---|---|
| | 97.5% | 98.0% | 98.5% | 99.0% | 99.5% | 99.9% |
| | Weekly Deliveries | | | | | |
| 150 | 8.2% | 8.5% | 8.8% | 9.2% | 9.4% | 9.2% |
| 450 | 7.4% | 7.8% | 7.8% | 8.2% | 8.6% | 8.8% |
| | Daily deliveries | | | | | |
| 150 | 8.0% | 8.4% | 8.8% | 9.1% | 9.4% | 9.3% |
| 450 | 5.9% | 6.3% | 6.7% | 7.0% | 7.3% | 7.3% |
| | Central Warehouse | | | | | |
| 150 | 9.4% | 9.8% | 10.1% | 10.4% | 10.6% | 10.6% |
| 450 | 7.4% | 7.7% | 8.0% | 8.3% | 8.5% | 8.5% |

Source: Own calculations.

**Table 6.** Margins in Different Strategies (Higher Value of Goods, Standard Deviation = 4.44 t).

| Distances [km] | Customer Service | | | | | |
|---|---|---|---|---|---|---|
| | 97.5% | 98.0% | 98.5% | 99.0% | 99.5% | 99.9% |
| | Weekly Deliveries | | | | | |
| 150 | 7.8% | 8.2% | 8.6% | 9.1% | 9.4% | 9.6% |
| 450 | 7.6% | 8.1% | 8.5% | 8.9% | 9.3% | 9.5% |
| | Daily deliveries | | | | | |
| 150 | 7.8% | 8.2% | 8.7% | 9.1% | 9.5% | 9.7% |
| 450 | 7.5% | 8.0% | 8.5% | 8.9% | 9.3% | 9.5% |
| | Central Warehouse | | | | | |
| 150 | 8.1% | 8.5% | 8.9% | 9.4% | 9.7% | 9.9% |
| 450 | 7.8% | 8.3% | 8.7% | 9.2% | 9.5% | 9.7% |

Source: Own calculations.

*4.3. Impact of the Distribution Strategy on the Internal Total Costs of Distribution*

The next stage of the analyses was to calculate the total costs of distribution, understood here as the costs of delivering goods to customers, capital costs of inventories (apart from warehousing costs) and the costs of lost sales.

The calculations were performed for each variant. The cases with the lowest costs were selected. The results for total internal costs are presented in Tables 7 and 8 and Figures 3 and 4. Both in the case of cheaper and more expensive products, the highest costs are generated in the strategy of daily deliveries to local warehouses in the decentralized strategy. For this reason, this strategy will be omitted in further analyzes.

The warehousing centralization strategy is profitable up to distances of 600 km for cheaper goods. However, in the case of more expensive goods, this strategy is profitable for distances up to 850 km. With the assumptions adopted here, one central warehouse could handle distribution, for example the whole of Poland and several neighboring countries. Therefore, the question of what factors influence the efficiency of this strategy at greater distances remains open.

**Table 7.** Impact of Distribution Strategies on Total Costs of Logistics (Lower Value of Goods, Standard Deviation = 4.44 t).

| Strategy | Weekly Deliveries | Daily Deliveries | Central Warehouse |
|---|---|---|---|
| Distance [km] | | PLN | |
| 350 | 26,540 | 29,469 | 22,755 |
| 450 | 28,794 | 33,229 | 26,554 |
| 550 | 31,049 | 36,990 | 30,353 |
| 650 | 33,303 | 40,750 | 34,151 |
| 750 | 35,558 | 44,510 | 37,950 |
| 850 | 37,813 | 48,271 | 41,749 |
| 950 | 40,067 | 52,031 | 45,547 |
| 1050 | 42,322 | 55,792 | 49,346 |
| 1150 | 44,576 | 59,552 | 53,145 |

Source: Own calculations.

**Table 8.** Impact of Distribution Strategies on Total Costs of Logistics (Higher Value of Goods, Standard Deviation = 4.44 t).

| Strategy | Weekly Deliveries | Daily Deliveries | Central Warehouse |
|---|---|---|---|
| Distance [km] | | PLN | |
| 350 | 56,194 | 53,199 | 48,016 |
| 450 | 58,460 | 56,976 | 51,825 |
| 550 | 60,725 | 60,754 | 55,634 |
| 650 | 62,991 | 64,532 | 59,443 |
| 750 | 65,257 | 68,310 | 63,252 |
| 850 | 67,523 | 72,088 | 67,061 |
| 950 | 69,789 | 75,865 | 70,870 |
| 1050 | 72,054 | 79,643 | 74,679 |
| 1150 | 74,320 | 83,421 | 78,488 |

Source: Own calculations.

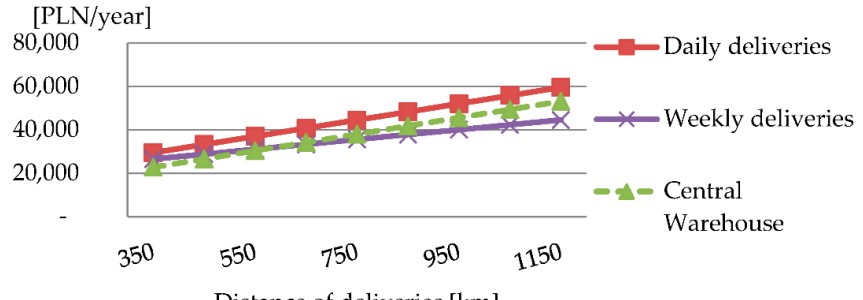

**Figure 3.** Impact of Distances on Total Costs of Distribution (Lower Value of Goods).

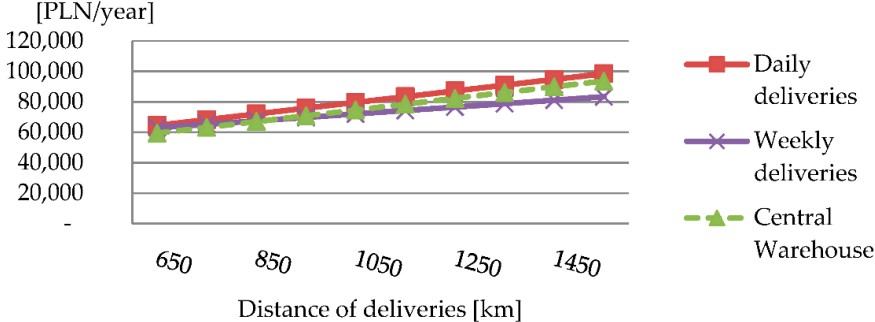

**Figure 4.** Impact of Distances of Deliveries on Total Costs of Distribution (Higher Value of Goods).

To investigate the impact of individual parameters on the total costs of logistics processes, simulations were carried out for a larger number of variants: Annual sales volume, demand volatility, value of distributed goods, number of warehouses in a decentralized system, and the width of the product range, costs of warehousing and maintaining stocks, and transport distances—to distribution centers (500 km, 1000 km, and 1500 km) and in the final (50 km, 150 km, and 250 km). The Monte Carlo methodology was applied, i.e., the demand for each combination of parameters was randomized (using the randomization function in the spreadsheet). One thousand three hundred simulations were carried out in this way.

The results may be surprising to some extent. The impact of parameters related to the product itself and the cost of inventories and storage is by far the smallest, although it does have some impact. Centralized distribution of both cheaper and more expensive goods can be profitable.

The most important factor seems to be the distance of the journey. With distances of up to 500 km to distribution centers, in the vast majority of cases, the total costs of the centralization strategy are lower than in the decentralized system, regardless of the type of goods and their sales characteristics. With longer transport distances up to 1000 km, and especially 1500 km, the profitability of the centralization strategy took place only in a few cases, even regardless of the distances at which the final distribution is carried out. An interesting regularity is that on such routes centralization is more profitable if in a decentralized system either the service level is lower or the annual sales volume is lower. Therefore, it can be hypothesized that the level of service seems to be one of the most important efficiency factors for a given distribution strategy.

What may seem surprising, in some cases, the decentralized system was cheaper even with a combination of distances—500 km to the centers by rail transport and 250 km in final distribution by road transport, and therefore more expensive. This happened with higher sales, which is worth emphasizing—cheaper goods, and those whose sales are subject to considerable fluctuations. Moreover, there are cases (which may confirm the above hypothesis) when the service level is higher (approximately 99% of the availability of goods), even with greater fluctuations in sales. The reason, as can only be guessed at this stage, is that economies of scale have an impact here, thanks to which the processes of warehousing and maintaining inventories may also be lower. At a larger scale, a trade-up effect may occur, i.e., the costs of logistics processes are lower and the level of service is higher [13,14].

The results would require a deeper analysis, but undoubtedly larger companies may be more effective, e.g., in cooperation with railway carriers, and such cases also take place in Poland, where delivery distances from the center of the country to regions on its outskirts may be just around 500 km.

However, with distances of 1000 km and 1500 km to the Centers, in the vast majority of cases, the centralized system ceases to be profitable even with a lower level of service in the decentralized system (95–97%). This means that it is cheaper to transport by rail, even if the delivery distances in the final distribution, i.e., by road, are 150 km and 250 km, which should reduce the effectiveness of this strategy. However, in a few cases centralized distribution was cheaper with the same parameters, except for one—when the sales volume was higher, which could be another argument for the economies of scale.

Regarding the number of warehouses and the number of products, there is some impact on the level of logistics costs, which can also be interpreted quite easily—a more complex system and greater variety mean higher costs. However, it does not affect the level of customer service. The influence of these parameters, however, should also be the subject of further research.

### 4.4. The Impact of the Distribution Strategy on the External Costs

The last stage of the analysis was to calculate external costs of these two strategies.

In connection with the policy of sustainable development and the idea of internalizing external costs, research has been undertaken for many years to estimate total costs, taking into account external costs [15–20].

There are different methodologies for calculating external costs and the levels of these costs are estimated according to them [21–26]. External costs can be counted for tkm or vehiclekm and both methodologies have their advantages and disadvantages. They depend on various factors, including the vehicle load capacity. What is also important is the level of these costs presented in the literature has changed over last years—the external costs are lower according to recent studies (Tables 9 and 10). Road transport is also becoming more and more "green" thanks to technical progress and modern technologies. That may mean that economical efficiency centralization of distribution, when road transport is used, is in fact higher, even when external costs are taken into considerations.

**Table 9.** Average External Costs of Rail and Road Transport According to Different Methodologies.

| Delft | | Ricardo | Delft | | Marco Polo | |
|---|---|---|---|---|---|---|
| [€ct/vkm] | | | [€/1000 tkm] | | | |
| Rail-electr. (295 tons) | Road-motorways | Road-motorways | Rail | Road | Rail | Road |
| 80.31 | 8.47 | 8.72 | 7.90 | 34.00 | 4.48 | 18.50 |

Source: Brons, Christidis, (2013); Delft, (2008); Ricardo, (2014); Lubieniecka-Kocoń, (2012).

**Table 10.** Average External Costs of Rail and Road Transport Used in Simulations.

| Delft | | Marco Polo | |
|---|---|---|---|
| [€/1000 tkm] | | | |
| Rail | Road | Rail | Road |
| 1.30 | 4.20 | 4.48 | 18.50 |

Source: Delft, (2019); Marco Polo, (2013).

The calculations include two variants of these costs—lower (Delft [22]) and higher costs per 1 tkm (Marco Polo [23]).

It was also assumed that the mileage utilization in direct road transport in centralized distribution is very high and increases with distance, and with longer routes it can be even over 90%. However, in rail transport it is 50% and also—because of shorter distance—in the "last mile" performed by road transport.

The total external costs a distribution strategy can be calculated with the use of following formula:

$$TCextd = TCextw + TCextt$$

where

TC*extd—Total external costs of a distribution strategy*
TC*extw—Total external costs of Warehousing*
TC*extt—Total external costs of transport, which can be calculated with the following formula:*

$$TCextt = TCexttm + TCexttf = U*(UCexttm/Umm + UCexttl/Uml)$$

where

TC*exttm—Total external costs of transport on main road*
TC*exttf—Total external costs of transport in final delivery*
UC*exttm—Unit external costs of transport on main road*
UC*exttl—Unit external costs of transport in final delivery*
U*mm—Utilization of mileage on a main road*
U*ml—Utilization of mileage in final delivery*

The results of calculations of external costs of transport are shown in Table 11 and Figures 5–8.

**Table 11.** Impact of Distribution Strategies on External Costs of Transport.

| External Costs of Transport | Rail | Road | Rail | Road |
|---|---|---|---|---|
| | **5.2** | **16.8** | **17.92** | **74** |
| | **[PLN/1000 tkm]** | | | |
| Distribution system | Weekly deliveries | Central Warehouse | Weekly deliveries | Central Warehouse |
| Distances on a main route [km] | "Last mile" 50 km | | | |
| 400 | 5840 | 8400 | 21,736 | 37,000 |
| 600 | 7920 | 11,859 | 28,904 | 52,235 |
| 800 | 10,000 | 14,933 | 36,072 | 65,778 |
| 1000 | 12,080 | 17,684 | 43,240 | 77,895 |
| Distances on a main route [km] | Last mile 200 km | | | |
| 400 | 10,880 | 8400 | 43,936 | 37,000 |
| 600 | 12,960 | 11,859 | 51,104 | 52,235 |
| 800 | 15,040 | 14,933 | 58,272 | 65,778 |
| 1000 | 17,120 | 17,684 | 65,440 | 77,895 |

Source: Own calculations based on data from: Delft, (2019) and Marco Polo, (2013).

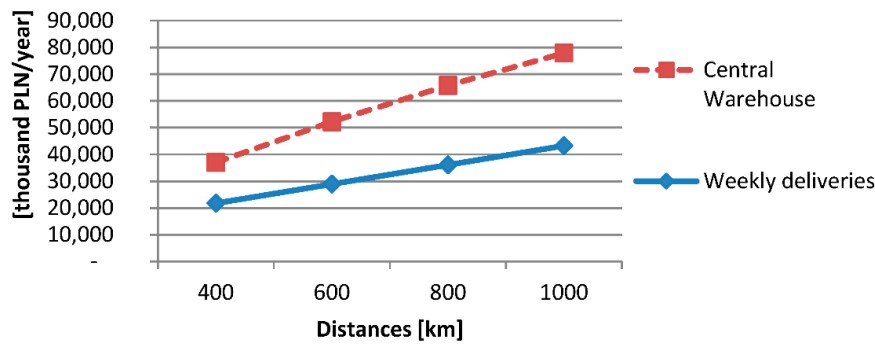

**Figure 5.** Impact of Distribution Strategies on External Costs (Lower ext. costs, last mile = 50 km).

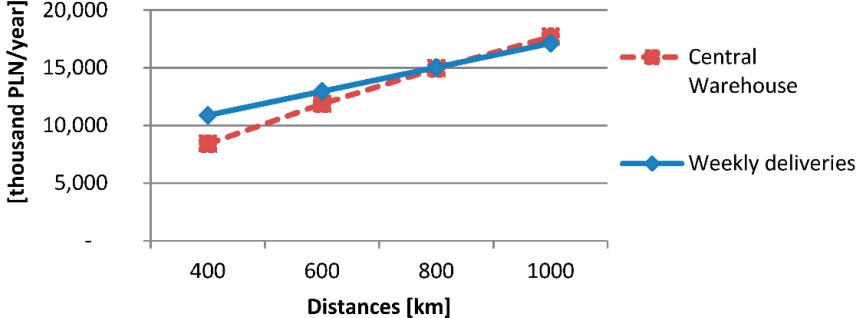

**Figure 6.** Impact of Distribution Strategies on External Costs (Higher ext. costs, last mile = 50 km).

**Figure 7.** Impact of Distribution Strategies on External Costs (Lower ext. Costs, last mile = 200 km).

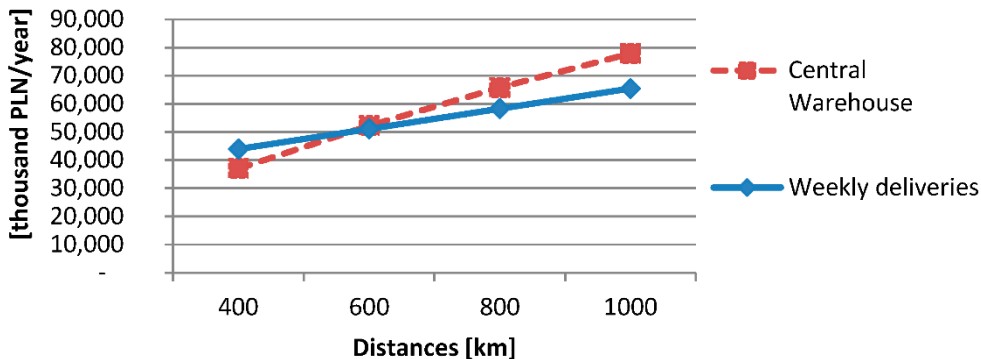

**Figure 8.** Impact of Distribution Strategies on External Costs (Higher ext. costs, last mile = 200 km).

For shorter distances (50 km) in final deliveries to customers the centralized distribution generates higher external costs than the decentralized (Figures 5 and 6) despite methodology of calculating these costs.

Methodology also plays an important role for longer "last miles" (200 km). The external costs deliveries to distribution centers in the decentralized system, although performed by the rail transport, are higher than in the centralized system even up to 800 km (Figures 7 and 8). When the Marco Polo methodology is applied, this distance is shortened to 600 km.

In order to assess the impact of the choice of a distribution strategy on the total external costs of distribution, however, the external costs in warehouses should also be calculated. Operations performed in warehouses also generate this kind of costs (movement of goods in warehouses, reloading operations). The level of these costs should be the subject of further research.

## 5. Discussion

The results of the simulations show that the centralization of warehousing can have a significant impact on the economic efficiency of an enterprise. The effects depend, however, on many different factors. The simulation method used by the author allows for accurate calculations (provided that the input data are reliable), but its disadvantage is the labor intensity associated with both the construction of this model and the simulation with its use. Undoubtedly, the use of a mathematical formula would be less laborious, but the author doubts that such a formula could be developed due to the high degree of complexity of the relations of logistic processes.

The simulation results do not support the benefits of using a centralization strategy over very long distances even for expensive goods—for example, serving Spanish market from a Central Warehouse in Rotterdam. Perhaps the costs of maintaining inventory and lost sales are actually lower in a centralized strategy. It should be borne in mind that the simulations were carried out using a spreadsheet in which the demand was randomly generated with a given probability and the assumption that it has a normal distribution. Perhaps the results would be different when based on data on the demand for products in specific companies.

In the centralized system, there may be a significant reduction in the level of stocks compared to the decentralized system. However, the results of the calculations did not confirm that the "square root" formula could apply, at least not in all cases. The square root formula is only applicable to some variants—namely high customer service and high sales fluctuations. In practice, as a result of erroneous demand forecasts, very large inventories may be created—larger than it would appear from the simulations carried out here, which are kept in local warehouses. Centralizing and increasing the flexibility of responding to current demand not only allows to reduce safety stocks, but also to eliminate the so-called excess stocks. This problem is faced by many companies that base their distribution strategies on stocks rather than on efficient and flexible deliveries.

The research results also confirm the correctness of the views that when considering the impact of centralization of warehousing on the level of stocks, all stocks and not only safety stocks should be taken into account (see Das [5] and Fleischmann [6]).

Choosing the right distribution strategy is associated with a typical optimization problem, which involves the so-called the effect of "trade-offs" between the costs of inventories and warehousing and transport, but also the costs of lost sales depending on the level of service. Many different factors affect the costs of logistics processes. The value of the goods and their selling price are important. The rates for logistics services (transport and storage) may vary. In the simulations presented here, the author relied on market information regarding rates for services, but they may even differ significantly from the prices for large customer. It is more difficult to estimate these costs if a given company has its own warehouses. In such a situation, not only the level, but also the structure, of these costs may be different. For services, variable costs are included in the calculation, and for own resources, mainly fixed costs.

The simulation assumes that the measure of logistic customer service is the availability of stock in the warehouse, which is a very important factor of a good service. However, extending the delivery route to the customer extends the time, which may also have an impact on the company's competitiveness. This aspect should also be the subject of further research.

The transport processes are probably more efficient than those adopted in the analyses presented here. As already mentioned, the centralization strategy is usually effective when the delivery is performed by specialized operators. They can efficiently deliver even small quantities of goods quickly and on time using their own resources. The centralization of inventories does not have to mean that goods are transported directly to customers over longer distances in smaller batches. The distribution network of these operators may be the substitute of the existing warehouse network. The goods will then only flow through, for example, a cross-docking system in order only to consolidate shipments. Thus, transport costs may be lower.

Similar comments can be made with regard to the external costs of transport. Research has been conducted in this area for years, but the results in the form of the level of these costs are very diverse. External costs depend not only on the mode of transport, but on different factors—e.g., the weight of the load and the degree of utilization of the vehicle capacity. The weight of a laden vehicle significantly affects fuel consumption, and therefore environmental pollution. It is probably wrong to average these costs, which should be estimated with high precision for specific cases—a given geographic area (or even a specific route), transport technology, and specific loads. Cost data found in various types of studies should probably also be regularly updated. In transport, we observe constant and large technological progress, which may be reflected in the increasing efficiency of means of transport in road transport and lower external costs of this mode of transport.

External costs generated in local and central warehouses should also be taken into account. It is estimated that the storage sector generates 25% of global warming emissions [9].

First of all, further research on external costs in various modes of transport should be carried out.

There are both different levels of these costs presented in the literature and different methodologies for calculating these costs. For example, in the Marco Polo methodology, external costs in rail transport are given for a train with a load of around 3000 tons (similar to Delft) and for road transport for loads of 7.7 tons. They are probably calculated for specific cases, i.e., the transport of bulk cargo by rail and the transport of consumer products (e.g., electronic consumer products) by road. It makes it difficult to compare external costs in different modes of transport for a given type of goods.

A separate issue is the usefulness of such simulations that take into account external costs for decision-making purposes. Some companies may be interested in the results of such research, at least for image purposes. For example, D.B. Schenker has a "Green Logistics" department, which calculates the level of external costs generated by logistics processes. Apart from the actual interest of entrepreneurs in social issues ("Corporate Social Responsibility"), they could at least potentially provide an opportunity to estimate the impact of the implemented logistics processes on external costs.

Research on the impact of the inventory centralization strategy on the broadly understood economic efficiency (including external costs) should be continued. There is a need to conduct empirical research in enterprises in order to investigate the influence of factors such as the size of the distribution network or the characteristics of distributed goods on economic efficiency, so that regularities in this area can be identified more precisely.

There is also a general problem related not only to the problem of optimizing the size of the distribution network, but also to the general problem of making decisions in the area of logistics. In the simulations used by the author, economic calculations were conducted, which hardly correspond to a traditional accounting. Some costs are not captured in the traditional accounting system. Few companies specify this type of cost in their reports [27–29].

The author hopes that the presented simulation results and conclusions that on this basis will be drawn will be verified by other researchers to confirm and specify these regularities.

**Funding:** This research received no external funding.

**Conflicts of Interest:** The author declares no conflict of interest.

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
