# Peer review of "Total Costs of Centralized and Decentralized Inventory Strategies—Including External Costs"

_sustainability, doi:10.3390/su12229346_

Round 1

Reviewer 1 Report

This version is much improved and much more clear than the original manuscript. Thank you for taking the time to make the improvements and to focus your writing on the readers and not just the research.

I do not have any major content or structure concerns or suggestions.

I do just have one question about the structure or presentation: In Table 2, why are 2 of the 6 scenarios identical? Both the 2nd and 3rd scenarios appear to be identical, with the same SD (2,4) and the same levels of customer service outcomes. If this is intentional, please explain what you are trying to show with this. If this is not intentional, please correct or remove 1 of the 2 identical scenarios. Please address Figure 2, as well, which shows the graphing of Table 2.

Beyond that, I think the story is good and the presentation is effective. The results are presented much better such that different readers can use different perspectives and apply them to their own situations or businesses.

There are a few spelling changes that need to be made - this may not be all of the changes, so be sure to perform a thorough spelling and grammar check: Row 90, Rows 121-123, Table 3 heading ("simulations"), Tables 5, 6 & 7 (be consistent with "stand. dev." and Row 404.

Author Response

Dear reviewer

  1. Two same columns in table 2: Thank you very much for noticing this error (my omission, of course, when inserting into the table). Now the chart looks even better.
  2. I corrected errors in the text and tables and made the term "standard deviation "

Thank you very much for your the review and valuable comments

Reviewer 2 Report

The author have made changes in the manuscript in accordance with the  requirements and suggestions.

The abstract is suggestive and highlights the objective and results of the research and the introduction was improved. Also, the research methodology is now clear presented.

The results of the research study are well grounded, but  I recommend to extend the comments for figure no 4,no 5,no 6, no7. Also, have been added new relevant references.

I consider that, after a hard work, the author have made the changes in alignment with recommendations.

Author Response

Dear reviewer

I completed the description of the calculation results in Tables 4-7 and explained which costs were calculated (total costs of distribution, which include costs of logistics processes and lost sales).

Thank you very much for your the review and valuable comments

Reviewer 3 Report

After improvement the paper can be published.

Author Response

Dear reviewer

Thank you very much for your the review and valuable comments

This manuscript is a resubmission of an earlier submission. The following is a list of the peer review reports and author responses from that submission.

Round 1

Reviewer 1 Report

Thank you very much for the opportunity of reading your work.
I found your research topic an interesting one with plenty of room for further development. Nevertheless, you have not yet identified an interesting research gap and your research design seems too simplistic given the complexity of the subject. At the same time, your option of using computer simulation data requires you to explore it in a more sophisticated way if you want to publish in a top journal such as Sustainability. Your result discussion is too technical and fails to inform theory in a significant way.

I encourage you to refine your work and pursue your efforts to produce meaningful research in this field.

Reviewer 2 Report

The subject is interesting, but there are a few numbers of issues that have to be reconsidered:

  1. The introduction is very abstract. The author does not present the level of exploration of the problem, there is not clear the novelty of the authors' research. I would suggest to structure the introduction by introducing the importance of the topic you analyzed, to summarize what we know about it and what are your objectives and expected contributions.
  2. The literature review is missing…. nothing about reliable findings of international researchers.
  3. Methods it should be deeper presented and explain why it was selected, which are the advantages and limits.
  4. Conclusions are rather general, there are missing the limits of research. 
  5. References needs improvement.

Work needs improvement and should be re-evaluated after correction.

Reviewer 3 Report

I would like to see considerable attention paid to determining the purpose of this study and the possible contribution to other researchers. Having read this manuscript multiple times, I do not know what the purpose is or what the contribution is.

There seems to be a fair amount of work that went into this study, and I appreciate that. The purpose appears to be to evaluate the costs - internal and external - of a logistics strategy based on a number of assumptions. However, I am concerned that such "evaluation" is not a significant purpose, especially given the enormous amount of static assumptions that need to be made within this evaluation. The conclusion - and, thus, possibly the contribution - is that there are many trade-offs to be considered in a logistics strategy. We already knew this.

There may be some practical implications of this work such that some distribution companies may learn something (e.g. companies with low value goods should and long transport distances should think twice before using a centralization strategy). But, I'll ask you: how many millions would you invest in any of the results estimated in this analysis?  Probably none, right? And that would make sense; everything done in this study is completely dependent on the assumptions and the context; as you explain in Section 4 "Discussion," the assumptions and context will differ for every company and their situations will not model your assumptions. Thus, you may not have any generalizable or actionable findings - for companies or for researchers.

A few more concerns / suggestions follow:

  1. My favorite part of the paper are the last 3 sentences in lines 401-406. This is where there is the greatest opportunity for a contribution - focusing on external costs. We know these change regularly; and we know these have changed significantly since the 2013 study that you use in your analysis. Focusing on external costs and using this as the connection to regulation or macro-policy can create contribution for future research and for policymakers. No rational company is going to voluntarily internalize these external costs unless they are forced to, by regulators or certain stakeholders. Yet this is where the potential for you to differentiate your study from a simple cost-benefit analysis with multiple assumptions.
  2. Who is the audience - researchers, regulators or corporate logistics managers? It's not clear from the study or the results. Perhaps if you focus on your specific audience, it can help you further revise your presentation and contribution.
  3. You can focus more on the trade-offs between short-term decisions and long-term dynamics. You do this, somewhat, by including lost sales in the model. Yet, as you explain in Section 4 "Discussion," you make assumptions about this that is likely disconnected from reality. What about marketing, image enhancement, public relations and brand reputation?  What about any companies that own their distribution facilities - how can this long-term investment be incorporated into the decision-making? How do external costs accumulate over the long-term? At what point does avoiding these costs benefit the company?
  4. Figure 1 is very helpful. Of course, is does lead us to think about the constraints and limitations of your design. We know you cannot include every possible assumption in your model, but it might be helpful to know why you chose the numbers you chose (12 products, 50km distance, 6 local warehouses) - how sensitive is the model to each of these assumptions? How can this be generalized for practical use by others?
  5. The other exhibits / graphs need to be cleaned up and improved. The axis scales are too large, the markers too large and it is difficult to differentiate between many of the lines.
  6. It appears that you are using a static / discrete sensitivity analysis approach to your modeling rather than a Monte Carlo simulation that more dynamically considers the distribution of each assumption. It likely doesn't make a difference for the extremes of your analysis, but a dynamic simulation might tell you more about the distribution of results in between the extremes.
  7. I know this is a clinical analysis of a simulation model and not new scholarly research, but I would like to see more of a review of the scholarly literature, highlight how this literature has evolved over the past 20 years or even 50 years as decentralization has changed. How has prior research - published in peer-reviewed journals - informed your design and your assumptions? And how do your findings advance that literature? This literature review should not cite your own work unless essential, but should learn from and build on what the entire community of scholarly work has done.

This is obviously a critical issue for companies, for regulators, for researchers and for us as citizens. We know there are many trade-offs related to this issue that affect many different stakeholders. However, simply saying there are many trade-offs and many unknowns is not enough of a contribution to be relevant or significant. We need to do better. Focusing on your purpose and refining your study design can possibly lead to generating that relevant contribution for researchers, companies and others.